# Impact of Financial Factors on the Economic Cycle Dynamics in Selected European Countries

**Bogdan Andrei Dumitrescu** [1,2,*] and **Robert-Adrian Grecu** [2]

1   "Victor Slăvescu" Centre for Financial and Monetary Research, Calea 13 Septembrie,
    050711 Bucharest, Romania
2   Department of Money and Banking, Faculty of Finance and Banking, Bucharest University of Economic
    Studies, 010961 Bucharest, Romania; robert.grecu@fin.ase.ro
*   Correspondence: bogdan.dumitrescu@fin.ase.ro

**Abstract:** The aim of this paper was to assess the impact generated by the financial market shocks on the economic cycle in European countries. In addition to the studies from the literature, which focus more on the developed economies, this paper also considered the situation at the level of a group of emerging economies to highlight the potential differences. In this sense, it was analyzed how the shocks at the level of the banking sector, those at the level of the capital market, and those at the level of the real estate market influence the dynamics of the economic cycle. Both econometric models for the individual analyses of each state, such as the Bayesian vector autoregression model, and models at the level of groups of states, such as panel regressions, were used for the period 2007–2022. The results showed a strong connection between the dynamics of the financial system and that of the real economy. In addition, the impact of financial factors on the economic cycle tends to be much stronger and more significant in the case of developing countries, compared to developed ones. In this regard, it was recommended that fiscal and monetary policies should be coordinated to generate the expected effect on the economy.

**Keywords:** economic cycle; shocks; financial factors

## 1. Introduction

The 2007–2008 global financial crisis (GFC) revealed that, in addition to the study of the economic cycle, a key element that must be considered by economic policymakers is the financial cycle. As could be observed during the GFC, the dynamics of the financial cycle exhibited very strong effects on the real economy, ultimately generating an economic recession.

The purpose of this paper was to identify how financial market shocks influence the economic cycle. This topic is of interest as the nature of the connection between the financial market and the real economy can have an important impact on the effectiveness of the economic policy measures implemented. Thus, the dynamics of the credit market, the capital market, or even the real estate market can scale, or depending on the case, they can massively reduce the effectiveness of economic policy measures.

The interaction between the financial and the economic cycle is more easily noticeable in periods of crisis, when the two types of cycles tend to synchronize, as the results of the study conducted by Haavio (2012) showed. However, it is important to analyze whether the connection between them is maintained in general and also during periods that are not characterized by economic tensions.

According to the studies in the dedicated literature, the dynamics of the financial cycle is the one that determines the dynamics of the economic cycle, two of the papers that reached that conclusion being Gómez-González et al. (2014) but also Shen et al. (2019). Given the above, this study considered that direction within the nexus between financial factors and the dynamics of the economic cycle.

As the experience of the last century has shown, financial crises can begin in different sectors of the financial market. Several crises started from the credit side, the excessive indebtedness of the households or non-financial companies being the generating element of the financial and economic repercussions. Other crises of a financial nature started due to excessive increases in real estate market prices, which ultimately generated a wide spectrum of negative effects at the level of the entire economy. Moreover, the third type of financial crisis started due to the capital market dynamics, with the high volatility leading to the accumulation of important vulnerabilities in the financial market. In this regard, the financial factors used in this study were selected to cover the entire spectrum of the financial system; more precisely, variables that characterize the activity of the banking sector, variables that characterize the dynamics of the capital market, and variables that characterize the dynamics of the real estate market were included.

To investigate whether there are differences in the direction and magnitude of this link, conditional on the level of development of the states, two groups of countries were analyzed. The first is a group consisting of five developing countries from Central and Eastern Europe, namely Romania, Czechia, Poland, Hungary, and Bulgaria, while the second is made up of five developed states from the central-western region of Europe, namely Germany, France, Austria, Spain, and Italy. Comparing the results obtained at the level of the two groups of states, as a general conclusion, the impact of financial factors on the economic cycle tends to be much stronger and more significant in the case of developing states compared to developed ones.

The novelty of this paper comes from enriching the scarce literature related to the link between financial factors and the economic cycle dynamics for developing countries. Different from other works in the literature, which carry out studies on developed economies such as that of the United States of America (Constantinescu and Nguyen (2021), Furlanetto et al. (2019), or Furlanetto et al. (2021)) or on other developed economies such as those in the G7 (de Winter et al. 2021), our paper analyzed the link between the economic cycle and financial factors at the level of the selected European developing countries. Also, in addition to articles in the literature that focus on the evaluation of credit market shocks on the economic cycle (e.g., Bartoletto et al. 2019 and Beltran et al. 2021), in our work, shocks from other financial market sectors, such as the real estate and capital markets, were considered.

In what follows, this paper includes the summary of the results from the correspondent literature on this topic, the description of the research methodology and data used, followed by the results and the conclusions drawn from our study.

## 2. Literature Review

The papers from the literature exhibit a significant link between the financial cycle and the economic cycle, with financial factors representing important determinants in economic dynamics. In general, the authors determined the economic cycle based on the gross domestic product, and the financial cycle based on indicators from the credit market, real estate market, or capital market.

The work of Karagol and Dogan (2021), which had Turkey as a case study, illustrated that there is a strong connection between the financial cycle and the economic cycle. In this sense, the authors recommend economic policymakers to consider the role that financial factors can have in the efficiency level of the measures adopted by them. In the case of this study, the proxy variables for the dynamics of the financial cycle were not only those regarding the dynamics of credit but also those regarding the dynamics of the capital market.

Important results were also presented in the study of Bartoletto et al. (2019), whose analysis was based on the Italian economy. In this case, the financial cycle was mainly identified through the dynamics of credit. The main results indicated that periods of economic recession associated with credit crunches are much more severe than periods of recession in which credit had no contribution. These authors also found that the response

of the economic cycle to credit shocks was much stronger during contraction periods of the financial cycle.

The paper of Berger et al. (2022) had the USA as a case study. In this work, the financial cycle was identified through credit dynamics and real estate price dynamics. The results of the analysis illustrated that the financial sector had an important role in the dynamics of the economic cycle during the Great Recession. Also, a financial shock can generate a negative correlation between the lagged credit cycle and the contemporaneous economic cycle. In this context, caution was suggested in associating periods of economic expansion with periods of expansion of the financial cycle.

Another approach in determining the cycles is that through the Beveridge–Nelson filter used by Morley and Wong (2020). In addition to the different filtering method, this study also employed Bayesian VAR-type models to identify macroeconomic variables that provide information about the dynamics of the economic cycle. Their results showed that, in addition to economic growth, variables such as the inflation rate but also the unemployment rate have an important impact on the dynamics of the economic cycle. Another work that expands the methodological framework for the analysis of economic and financial cycles is Bulligan et al. (2019), whose study was based on the Italian economy and used multivariate filters to identify the trend and the cyclical components.

The study carried out by Schuller (2020) showed that the analysis of the financial cycle only through credit indicators (e.g., credit to GDP gap) can lead to the false identification of periods of expansion or contraction. In this perspective, it is useful to consider other options of identifying the phases of financial cycles, starting from variables regarding other sectors of the financial market, but also additional filtering methods compared to the standard ones recommended by the BCBS (Basel Committee on Banking Supervision). However, there are also works, such as Beltran et al. (2021), which show that the method of calculating the financial cycle based on credit dynamics, or through the credit to GDP gap indicator, can be optimized to improve its ability to identify the phases of the financial cycle.

Another work, by de Winter et al. (2021), carried out their analysis of the economic and financial cycles at the level of eight developed countries, specifically the seven countries of the G7 (the USA, UK, Japan, Canada, Germany, France, and Italy) but also the Netherlands. The results of this analysis showed that, at the level of all eight economies, there is a high degree of co-movement of the financial cycle, identified through real estate prices, and the economic cycle identified through gross domestic product.

There are also studies that have analyzed the link between the economic cycle and financial factors for a long period of time. In this sense, the study conducted by Constantinescu and Nguyen (2021) can be mentioned, which analyzed the American economy for a period of more than a century. The results of this study confirmed the essential role of financial factors, such as credit, the prices of financial assets, or the real estate market, in the dynamics of the economic cycle. Another study of interest at the level of the USA economy is that of Furlanetto et al. (2019); its results showed that financial shocks represent an important determinant of the gross domestic product dynamics (of investments) but have a limited effect on the dynamics of inflation. The financial shocks identified via the real estate market prices are the ones that play the most important role in the dynamics of the gross domestic product. Another study (Furlanetto et al. 2021) revealed that financial frictions and financial market shocks not only have an important impact on the dynamics of the economic cycle but also on how other determinants influence economic fluctuations.

An important paper analyzing the impact of financial factors on the economic cycle is represented by Christiano et al. (2010). The methodology used was based on a DSGE (dynamic stochastic general equilibrium) model applied at the level of the Eurozone states but also at the level of the USA (United States of America). The results of this study showed that shocks at the level of the financial system are important determinants of economic fluctuations.

Another reference work on this topic is the one of Iacoviello (2015), whose results showed that the impact of financial shocks affecting highly indebted sectors of the economy

accounts for two-thirds of the output collapse during the Great Recession. In their analysis, a DSGE model was used, in which a recession starts as a result of the losses suffered by the banks and their inability to provide enough credit to the real economy. Similarly, Borio et al. (2016) showed that financial factors play a key role in explaining the dynamics of the economic cycle.

Other papers have shown that there is Granger causality between the financial cycle and the economic cycle. In this respect, the study of Sala-Rios et al. (2016), whose analysis was based on data for Spain, showed that the economic cycle was caused by the financial cycle. The financial and economic cycles were determined by means of the Hodrick–Prescott filter, a similar procedure being employed in the present paper.

Another work that studied the connection between financial factors and the economic cycle through causality is that of Aravalath (2020), which used India as its case study for the period 1990–2019. The results also showed that the financial cycle is the one that causes the dynamics of the economic cycle. Another work that reached similar conclusions is that of Gómez-González et al. (2014), with their study having been carried out at the level of a group of South American states.

The link between financial factors and the real economy may differ depending on the phases of the economic cycle. In this sense, in the work of Antonakakis et al. (2015), a study was carried out at the level of the G7 (Group of Seven) states, which showed that the connection between the financial and the economic cycles becomes much closer in periods of recession, compared to periods of economic expansion.

Another work that showed a strong correlation between financial factors, materialized in the financial cycle, and the economic cycle is that of Akar (2016). The analysis was carried out at the level of Turkey for the period 1998–2014, and the results showed that there is a strong and positive correlation between the financial cycle and the economic cycle.

Another perspective of interest is the one regarding the impact of financial factors on the economic cycle, conditioned by the level of development of a state/province. The work of Shen et al. (2019), whose analysis deals with China's situation, showed that, in general, the phases of the financial cycle are the ones that determine the dynamics of the phases of the economic cycle. This result was recorded with a higher frequency and intensity within the richest provinces in the sample.

Another important perspective on the link between financial factors and the economic cycle concerns the period in which the effects of shocks become visible. An analysis at the level of the state of Denmark, carried out in the paper by Grinderslev et al. (2017), showed that the synchronization of the financial cycle with the economic one is felt more strongly in the medium term. Another paper that illustrated a stronger connection between the two types of cycles, in the medium term, is that of Škare and Porada-Rochoń (2020), whose study was based on data for the United Kingdom, using an econometric methodology based on spectral Granger causality. In the same area, a similar result was also discovered for the USA economy by Yan and Huang (2020) using wavelet functions but also VAR-type (vector autoregression) models.

Berger et al. (2020) analyzed the effect of financial factors on the economic cycle by means of Bayesian VAR models. The results of their study showed that the impact of financial factors on the output gap was much stronger in the period after 2000 than in the period before this year. Regarding the direction of influence, the financial cycle shock generates a positive contemporaneous impact on the output gap, but it becomes negative if the lagged effect of this shock is quantified.

However, it should be mentioned that there are also works whose results show a low connection between financial factors and the economic cycle, an example in this sense is the work by Apostoaie and Percic (2014), which was carried out based on data for 20 European states. The results of their study showed that, at the level of the analyzed states, there was no causality between the financial cycle and the economic cycle. Other studies have shown that the connection between financial factors and the economic cycle is limited; an example

of this is the work of Ahmad and Sehgal (2018), the authors of which studied a group of countries in the South Asian area by means of dynamic spillover models.

## 3. Methodology and Data

The objective of this paper was to identify how financial factors influence the economic cycle. In this sense, 2 groups of states were analyzed; the first was a group consisting of 5 developing countries from Central and Eastern Europe, namely Romania, Czechia, Poland, Hungary, and Bulgaria, while the second was made up of 5 developed states from the central-western region of Europe, namely Germany, France, Austria, Spain, and Italy. Data with a quarterly frequency were used, and the period analyzed was between Q4 2007 and Q4 2022.

The financial factors used in this study were selected to cover the entire spectrum of the financial system, more precisely, the activity of the banking sector, the dynamics of the capital market, and the dynamics of the real estate market. To characterize the activity of the banking sector, the dynamics of lending were used for both the household sector and for the sector of non-financial companies. To characterize the dynamics of the capital market, the CLIFS (Country-Level Index of Financial Stress) indicator was used as a proxy, and to characterize the dynamics of the real estate market, a proxy was used for the dynamics of residential property prices (RRE). In addition to financial factors, control variables were also used in the panel data models. In this sense, the harmonized inflation rate was used as well as the dynamics of the stock of private investments at the level of each state.

Credit data were taken from the ECB (European Central Bank) database in nominal quarterly values. For each quarter, the rate of change was calculated, considering the dynamics of the variable in comparison with the same quarter of the previous year, to prevent the non-stationarity problem. In this way, the data series included in the model were the growth rates of lending at the level of the two sectors, households, and non-financial companies. CLIFS data were taken in quarterly format from the ECB database and were used in the analysis in the same way. Regarding the RRE prices, the data were taken in quarterly format from the ECB database directly in the form of percentage change. The inflation rate was taken in the quarterly format directly in the percentage form from the Eurostat database. In the case of private investments, the data were taken in quarterly nominal format from the Eurostat database and were transformed into growth rates based on the dynamics of the variable in comparison with the same quarter from the previous year. Having all the datasets in percentage form, after the proper transformation, helped us to prevent potential econometric issues.

The dependent variable, the economic cycle, was calculated by means of the Hodrick–Prescott filter, the model being described in detail in the following paragraphs, similarly to the procedures applied in the dedicated literature. The input data on nominal gross domestic product, which were later passed through the logarithm operator and filtered, were taken from the Eurostat database. The economic cycle was calculated for the same period with the rest of variables used in the B-VAR (Bayesian VAR) model, specifically between Q4 2007 and Q4 2022.

The econometric programs used for data preprocessing were Eviews and Matlab, and an extensive presentation of the econometric models used has been made below.

In the empirical analysis, in accordance with relevant works, such as Tamási and Világi (2011) or Berger et al. (2020), two types of econometric models were used. On the one hand, Bayesian VAR (B-VAR) models were used to identify the effect caused by shocks in the financial system at the level of each state individually; on the other hand, panel data models were used to identify whether there were significant differences in impact caused by the level of development of the states.

To determine the economic cycle, a Hodrick–Prescott type filter was used. This filter has the property of decomposing a data series ($x_t$) into two important components: the

trend and the cyclical component (Formula (1)):

$$x_t = \tau_t + c_t \tag{1}$$

where $\tau_t$ represents the trend of the variable ($x_t$), and $c_t$ represents its cyclical component. To achieve this decomposition, the loss function presented in Formula (2) was minimized:

$$\sum_{t=1}^{T}(x_t - \tau_t)^2 + \lambda \sum_{2}^{T-1}\left[(\tau_{t+1} - \tau_t) - (\tau_t - \tau_{t-1})\right]^2 \tag{2}$$

where $\lambda$ represents the smoothing parameter. To determine the economic cycle, this parameter was calibrated according to the work of Hodrick and Prescott (1981) to the value of 1.600. This value was determined based on Formula (3):

$$\lambda = \frac{\sigma^2(c_t)}{\sigma^2[(\tau_{t+1} - \tau_{t-1})]} \tag{3}$$

After determining the economic cycle, the first type of econometric model was applied, more precisely B-VAR. Different from the classical approach, Bayesian econometrics treats each parameter as a random variable following a distribution, according to Bayes' rule (Formula (4)):

$$\pi(\beta|y) \propto f(y|\beta)\pi(\beta) \tag{4}$$

where $\pi(\beta|y)$ represents the posterior distribution of the parameters conditional on the data, $\pi(\beta)$ represents the prior distribution of the parameters, and $f(y|\beta)$ represents the likelihood function. Thus, an important step of this type of analysis is the selection of a priori distributions of the parameters, with Minnesota-type priors being used in this study. In this case, the likelihood function is represented in Formula (5):

$$f(y|\beta) \propto exp\left[-\frac{1}{2}(y - \overline{X}\beta)'\overline{\Sigma}^{-1}(y - \overline{X}\beta)\right] \tag{5}$$

The a priori distribution of the vector of coefficients ($\beta$) is presented in Formula (6):

$$\pi(\beta) \propto exp\left[-\frac{1}{2}(\beta - \beta_0)'\Omega_0^{-1}(\beta - \beta_0)\right] \tag{6}$$

Following the substitutions, the posterior distribution of the parameters conditional on the data takes the form:

$$\pi(\beta|y) \propto exp\left[-\frac{1}{2}(y - \overline{X}\beta)'\overline{\Sigma}^{-1}(y - \overline{X}\beta)\right] \times exp\left[-\frac{1}{2}(\beta - \beta_0)'\Omega_0^{-1}(\beta - \beta_0)\right] \tag{7}$$

A comprehensive overview of the demonstrations is provided in Dieppe et al. (2016). The BEAR toolbox (Bayesian estimation, analysis, and regression) available for the Matlab program was used to operate B-VAR models.

The second type of model used, for the analysis at the level of the group of developing countries compared to that of developed countries, was panel regression. The mathematical form of the model used is presented in Formula (8):

$$OG_{i,t} = \gamma_1 + \gamma_2 GR\_L\_HH_{i,t} + \gamma_3 GR\_L\_NFC_{i,t} + \gamma_4 CLIFS_{i,t} + \gamma_5 RRE_{i,t} + \gamma_6 HICP_{i,t} + \gamma_7 GR\_PI_{i,t} + \varepsilon_{i,t} \tag{8}$$

where $OG_{i,t}$ represents the output gap in quarter $t$ within state $i$; $GR\_L\_HH_{i,t}$ represents the quarterly growth rate of credit granted to households in quarter $t$ within state $i$; $GR\_L\_NFC_{i,t}$ represents the quarterly growth rate of credit granted to non-financial companies in quarter $t$ within state $i$; $CLIFS_{i,t}$ represents the CLIFS indicator in quarter $t$ within state $i$; $RRE_{i,t}$ represents the growth rate of residential real estate prices in quarter $t$

within state *i*; $HICP_{i,t}$ represents the harmonized inflation rate in quarter *t* within state *i*; and $GR\_PI_{i,t}$ represents the private investment growth rate in quarter *t* within state *i*.

To identify the most suitable type of model to be used, the one with random effects or the one with fixed effects, the Hausmann test was employed in the case of both groups of states.

## 4. Results and Discussion

Regarding the econometric models, two approaches were used; in a first phase, the impact of the shocks was analyzed at the level of each state through B-VAR-type models, after which, to increase the level of robustness of the results, regression models with panel data were used.

In the case of B-VAR models, the first type of shock analyzed is the one originating from the banking sector, through the credit granted to non-financial companies. The results for the group of countries in the CEE (Central and Eastern European) region are presented in Figure 1.

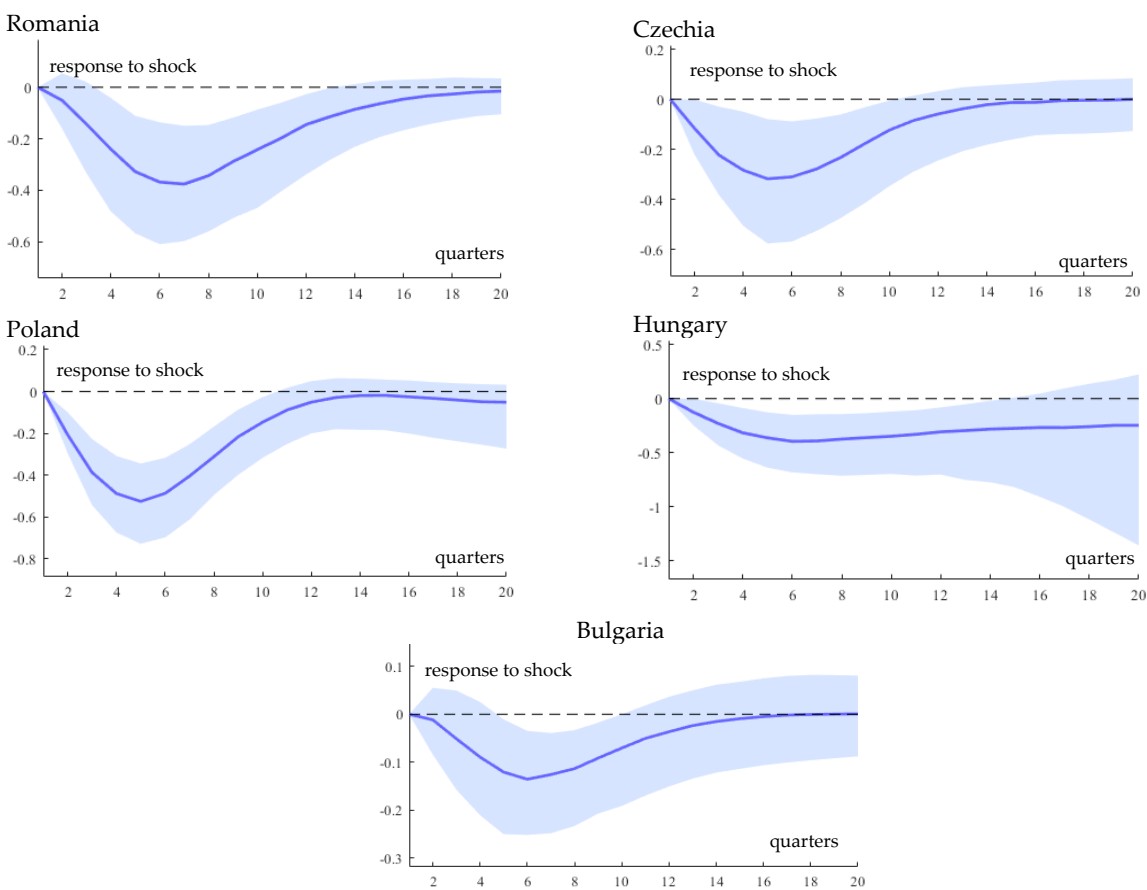

**Figure 1.** Dynamics of the economic cycle due to the shock on the level of loans granted to non-financial companies in the CEE states. Note: the line represents the response of the economic cycle due to the shock, and the highlighted area represents the 95% confidence interval.

In all five analyzed states (Figure 1), a shock in the level of credit granted to non-financial companies has the effect of reducing the output gap. Thus, if the economic cycle is in an expansion phase, the banking sector shock reduces its amplitude, while if the economy is in a recession phase, the banking sector shock intensifies the negative impact on the economy. A special case is that of Hungary, where, unlike the rest of the states, the impact of the credit shock persisted even after 20 quarters. One of the reasons that could justify this dynamic could be related to the relatively high size of the national banking sector of this state, measured via total domestic banking assets in GDP, at the level of

the CEE region. Another reason may be related to the relatively high level of financial intermediation compared to the rest of the analyzed CEE countries.

However, this type of effect is not conditioned via the level of development of the states, as in the case of most developed countries in the central-western region of Europe, analyzed in this paper, the same type of phenomenon was observed (Appendix A). However, special cases regarding this dynamic have been recorded in Spain and France, where the impact of such a shock is relatively low. This was caused by the size of the financial sector of the two countries, which also includes a well-developed capital market. Thus, financial intermediation can be carried out both through banks and through the capital market.

In addition to the shock that occurred at the level of lending to non-financial companies, the impact of a shock that occurred at the level of lending to household segments was also analyzed. In the case of the CEE countries, a shock to the crediting of this sector has the effect, in most states, of a decrease in the output gap (Appendix B). A special case was that of Hungary, where the shock remained persistent even after 20 quarters from the moment of its appearance. This result can also be justified via the relatively high level of financial intermediation in this country, compared to the rest of the states in the region. In the case of most developed states in the central-western region of Europe, the effect is one opposite than for the developing ones (Appendix C). A factor that could argue for this difference is related to the importance of lending to this segment in developed countries, compared to the situation in developing countries. Moreover, the only developed country where a more pronounced response of the economic cycle dynamics was observed, due to the shock, was Spain; however, in this case, the response was also not statistically significant.

The second type of shock analyzed was the one originating from the capital market, in this case, materialized at the level of the financial stress index (CLIFS). The results for the CEE group of states are presented in Figure 2.

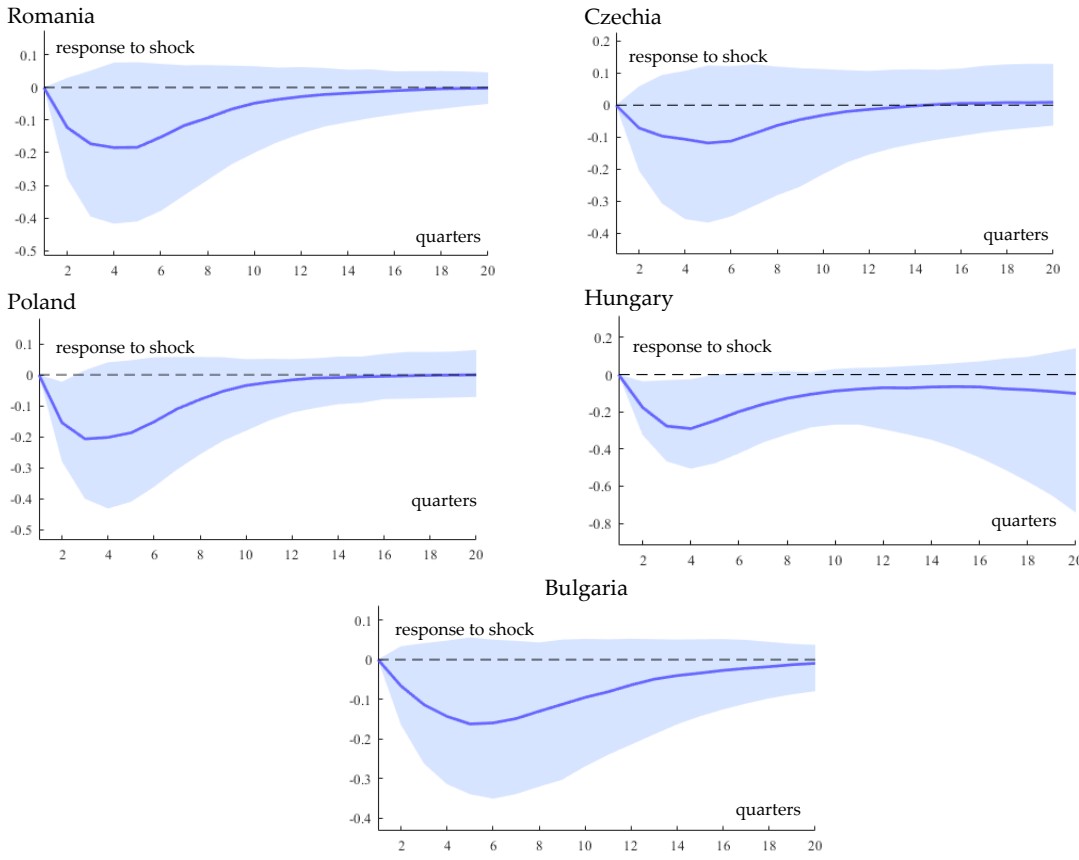

**Figure 2.** Dynamics of the economic cycle due to the CLIFS shock in the CEE states. Note: the line represents the response of the economic cycle due to the shock, and the highlighted area represents the 95% confidence interval.

A shock at the level of the financial stress index had the effect of reducing the output gap in all five analyzed CEE states. In this sense, during periods of expansion, the shock on the capital market diminishes the amplitude of the economic cycle, whereas during periods of economic recession, the negative impact on the economy is intensified. In this case, a similar trend was observed at the level of all five analyzed developing states. However, this type of effect is not conditioned by the level of development of the states, as in the case of most developed states in the central-western region of Europe, analyzed in this paper, the same type of phenomenon was observed (Appendix D). Also, in this case, a similar trend was observed at the level of all five analyzed developed states.

The third type of shock analyzed was the one originating from the real estate market. The results for the group of CEE states are presented in Figure 3.

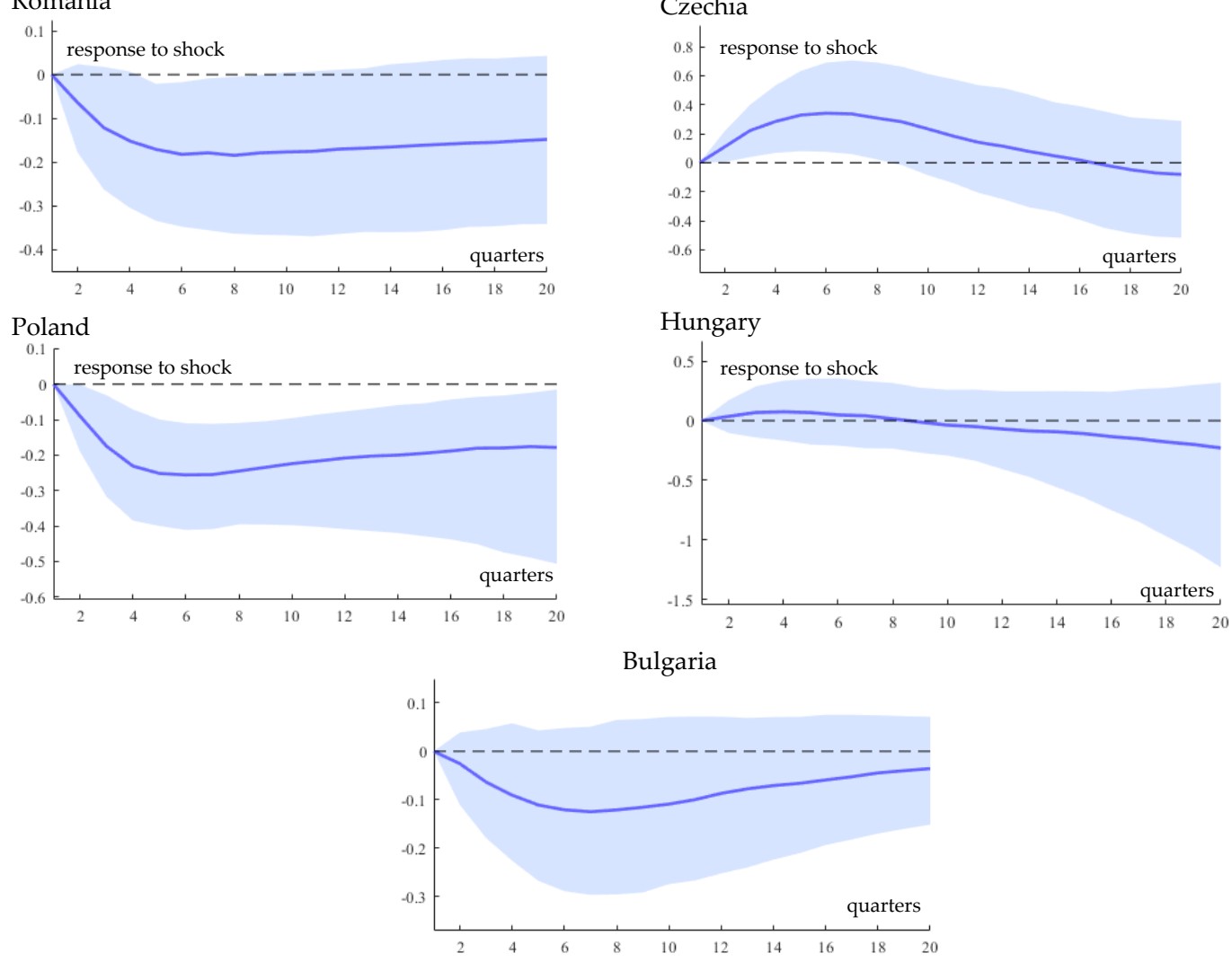

**Figure 3.** Economic cycle dynamics due to the real estate price shock in CEE states. Note: the line represents the response of the economic cycle due to the shock, and the highlighted area represents the 95% confidence interval.

A real estate price shock exhibits a distinct effect for different CEE states. Thus, if in the case of Romania, Poland, and Bulgaria, the shock materialized in this area generates a decrease in the output gap, in the case of Czechia, the effect is one in total opposition. The explanation of the result for Czechia could be provided by a much higher degree of synchronization of the financial cycle with the economic one in this state. Moreover,

considering these results, the real estate market in this country has a more pronounced impact on the macroeconomic framework, compared to the rest of the states in the region.

In the case of the developed states in the central-western region of Europe, for this type of shock, the effects on the output gap were closer, from the perspective of the direction of influence, to those recorded in Czechia (Appendix E). A special case was registered in Italy, Spain, and France, where real estate prices have a very low influence on the economic cycle dynamics. A reason for this result could be that other factors have a more pronounced impact on the general economic dynamics of these states.

The second method of analysis was carried out in an aggregated manner at the level of the groups of states analyzed by means of models with panel data. The first representation is that of the results for the panel of CEE states (Table 1).

**Table 1.** Econometric results at the level of the CEE group of states.

| Regressor | I | II | III | IV |
| --- | --- | --- | --- | --- |
| CLIFS | −0.0203 (0.0159) | −0.0267 (0.016) * | −0.0284 (0.0157) * | −0.0272 (0.0156) * |
| HH_L | | −0.0448 (0.0201) ** | −0.0027 (0.0235) | 0.0193 (0.0251) |
| NFC_L | | | −0.0796 (0.0242) *** | −0.0835 (0.024) *** |
| RRE | | | | −0.0255 (0.0109) ** |
| INFL | 0.0575 (0.078) | 0.0495 (0.0774) | 0.1386 (0.0806) * | 0.186 (0.0824) ** |
| PR_INV | 0.0435 (0.0168) ** | 0.0547 (0.0174) *** | 0.0647 (0.0173) *** | 0.0629 (0.0171) *** |

Note: *—the parameter is statistically significant at the 10% level; **—the parameter is statistically significant at the 5% level; and ***—the parameter is statistically significant at the 1% level. For each of the variables, the estimated parameter is noted, and the standard error level is noted in parentheses. Source of the data: own processing based on data provided by Eurostat and ECB.

In the case of this group of states, four panel regression models were run, in which different financial factors were integrated one by one sequentially. The model whose results were the most significant was IV, in which all the financial factors covered in the analysis were included. Thus, it is observed that an increase in the level of financial stress, materialized via an increase in the CLIFS, has a decreasing effect on the level of the output gap. A similar result was obtained individually for each of the states in this group through B-VAR models.

Another significant result is related to the impact of the dynamics of lending to the segment of non-financial companies. In this sense, an increase in lending to this sector has the general effect of a decrease in the output gap. However, it should be noted that only the increase in lending to the segment of non-financial companies had this effect; the dynamics of lending to the household sector did not significantly influence the dynamics of the economic cycle at the level of this group of states.

The third area analyzed was the impact of the dynamics of the real estate market on the economic cycle. In this sense, an increase in real estate market prices has the effect of a decrease in the output gap. Thus, if the economic cycle is in an expansion phase, the housing market shock reduces its amplitude, while if the economy is in a recession phase, the banking sector shock intensifies the negative impact on the economy.

The second group of states analyzed was that of developed countries from the central-western region of Europe (Table 2).

Similarly, to the situation at the level of the CEE group of states, and within the group of developed states, an increase in the CLIFS has the effect of a reduction at the level of the output gap. However, it should be noted that within this group of states, the impact of this factor is not only much lower in magnitude but also in terms of statistical significance.

**Table 2.** Econometric results at the level of the group of developed countries.

| Regressor | I | II | III | IV |
|---|---|---|---|---|
| CLIFS | −0.0146 (0.0123) | −0.0142 (0.0123) | −0.0129 (0.0121) | −0.0129 (0.0121) |
| HH_L | | −0.0148 (0.0241) | 0.0602 (0.0321) * | 0.062 (0.0348) * |
| NFC_L | | | −0.0905 (0.0263) *** | −0.0906 (0.0263) *** |
| RRE | | | | −0.0009 (0.0069) |
| INFL | −0.0216 (0.0803) | −0.0137 (0.0815) | 0.0142 (0.0801) | 0.016 (0.0814) |
| PR_INV | 0.1430 (0.0151) *** | 0.144 (0.0152) *** | 0.1484 (0.0149) *** | 0.1474 (0.0167) *** |

Note: *—the parameter is statistically significant at the 10% level; and ***—the parameter is statistically significant at the 1% level. For each of the variables, the estimated parameter is noted, and the standard error level is noted in parentheses. Source of the data: own processing based on data provided by Eurostat and ECB.

Regarding the link between the real economy and the banking sector, similarly to the situation in the CEE states, an increase in lending to the non-financial company segment results in a significant reduction in the output gap. In terms of magnitude, the impact of this factor tended to be similar for the two groups of states analyzed.

However, when the impact of the dynamics of credit granted to the households was analyzed, a difference was observed between the two groups of states. If in the case of the CEE states the impact of this factor was insignificant from a statistical point of view, in the case of developed states it is a significant one. The direction of influence was the same; however, it appeared that at the level of developed states, the dynamics of lending to the household segment have a more pronounced impact on the economic cycle.

Regarding the dynamics of the real estate market, the direction of influence is similar in the case of the group of developed countries to that obtained within the group of developing countries. Thus, an elevation in residential real estate market prices results in a decrease in the output gap. However, it should be noted that distinct from the case of CEE states, in the group of developed states the impact of real estate prices on the economic cycle was much lower, even insignificant from a statistical point of view.

Looking comparatively at the results obtained at the level of these two groups of states, as a general conclusion, the impact of financial factors on the economic cycle tends to be much stronger and more significant in the case of developing states, compared to already developed states. One of the arguments supporting this conclusion is related to the level of stability of the economy at the level of the two groups of states. The economies of developing countries are generally much more vulnerable to shocks, whether they are economic or financial, which is why the impact of any dynamic in the financial market is felt more strongly at their level.

Another argument that could support these conclusions is related to the level of development and complexity of the financial system in these two groups of states. A shock arising from the capital market, from the banking sector, or from the real estate market can be absorbed more easily under the conditions of a stable financial system so that the economic effects are diminished. This, on the other hand, is not possible in the same way in the case of states whose financial systems are poorly developed, and whose financial activity depends, to a very large extent, on only one area of the financial system.

## 5. Conclusions

The main aspect analyzed in this paper was related to the way in which the shocks in the financial system influence the dynamics of the economic cycle. More specifically, we have taken into consideration shocks that come from the banking system through the credit channel, those that come from the capital market and whose effect is to increase the financial stress index, and those that come from the real estate market through residential property prices. At the same time, this study was carried out both at the individual level

of a state and aggregated for the groups of states in CEE and those in the central-western region of Europe.

The papers from the literature mostly showed a significant link between the financial cycle and the economic cycle in developed countries, with financial factors representing important determinants in economic dynamics. In general, the most important financial factors with an impact on the economic cycle dynamics are the ones regarding lending, real estate prices, and the capital market.

The main results of the B-VAR analysis showed that a shock in the level of loans granted to non-financial companies has the effect of a decrease in the output gap in all five CEE states, which are in line with Berger et al. (2022). However, this type of effect was not conditioned by the level of development of the states, as in the case of most developed states in the central-western region of Europe, analyzed in this work, the same type of phenomenon was observed. This result is in line with other studies, such as Constantinescu and Nguyen (2021) and Furlanetto et al. (2021), who have shown that the indicators regarding lending have an important and significant impact on the dynamics of the economic cycle.

Regarding the capital market, a shock at the level of the financial stress index not only had the effect of reducing the output gap in all five CEE states analyzed but also in most of the developed states in the central-western region of Europe. This result is in line with Karagol and Dogan (2021), who identified a significant link between the capital market and economic cycle dynamics. Regarding the housing market, a shock to residential property prices had a distinct effect for different CEE countries in comparison with the developed ones. Thus, in the case of most developing countries, the shock materialized in this area generates a decrease in the output gap; in the case of most developed countries, the effect is one in total opposition. The significant impact of the real estate market dynamics on the economic cycle was also identified in other studies, including Furlanetto et al. (2019) and de Winter et al. (2021).

The results from the panel data model showed that, at the CEE level, the dynamics of the CLIFS indicator, the dynamics of lending to non-financial companies, and the dynamics of residential property prices significantly influence the dynamics of the economic cycle. Within the panel of developed states from the central-western part of Europe, only the dynamics of lending significantly influenced the dynamics of the economic cycle. These results are also in line with Bartoletto et al. (2019), who identified a strong link between lending and the economic cycle.

The practical implications of these results could be useful for economic policymakers, as any measure of monetary policy or fiscal policy that has effects on the activity of the banking sector, on the capital market, or on the dynamics of the activity of the real estate market influences indirectly and the dynamics of the economic cycle at the level of the respective state. In this sense, fiscal and monetary policies should be coordinated to generate the expected effect on the economy. On the contrary, if the direction of influence of the two types of policies is different, one of them being anti-cyclical while the other is procyclical, the level of efficiency of the implemented measures could be significantly reduced.

This analysis can be expanded, in future research, both from the perspective of the countries analyzed and the types of econometric models used. In terms of countries, a future study could consider all EU member states to have a complete picture of how financial factors influence the real economy at the European level. In terms of econometric models, future studies could not only extend the analysis using other methods of filtering and identifying business cycles but also using other financial factors than those included in the analysis to illustrate the interactions between the financial market and the business cycle. Regarding the limitations of this study, we can mention the relatively short period of time analyzed, due to the lack of other historical data available, and the lack of data for other developing states from the CEE region.

**Author Contributions:** Conceptualization, B.A.D. and R.-A.G.; methodology, B.A.D. and R.-A.G.; software, B.A.D. and R.-A.G.; validation, B.A.D. and R.-A.G.; formal analysis, B.A.D. and R.-A.G.; investigation, B.A.D. and R.-A.G.; resources, B.A.D. and R.-A.G.; data curation, B.A.D. and R.-A.G.; writing—original draft preparation, B.A.D. and R.-A.G.; writing—review and editing, B.A.D. and R.-A.G.; visualization, B.A.D. and R.-A.G.; supervision, B.A.D.; project administration, B.A.D.; funding acquisition, B.A.D. All authors have read and agreed to the published version of the manuscript.

**Funding:** This paper was funded through the research project "Analyzing uncertainties with respect to the Forecast of the Evolution of the Economic Environment in the context of recent global socio-economic shocks", within the Bucharest University of Economic Studies.

**Data Availability Statement:** The data presented in this study are available on request from the corresponding author.

**Conflicts of Interest:** The authors declare no conflict of interest.

## Appendix A

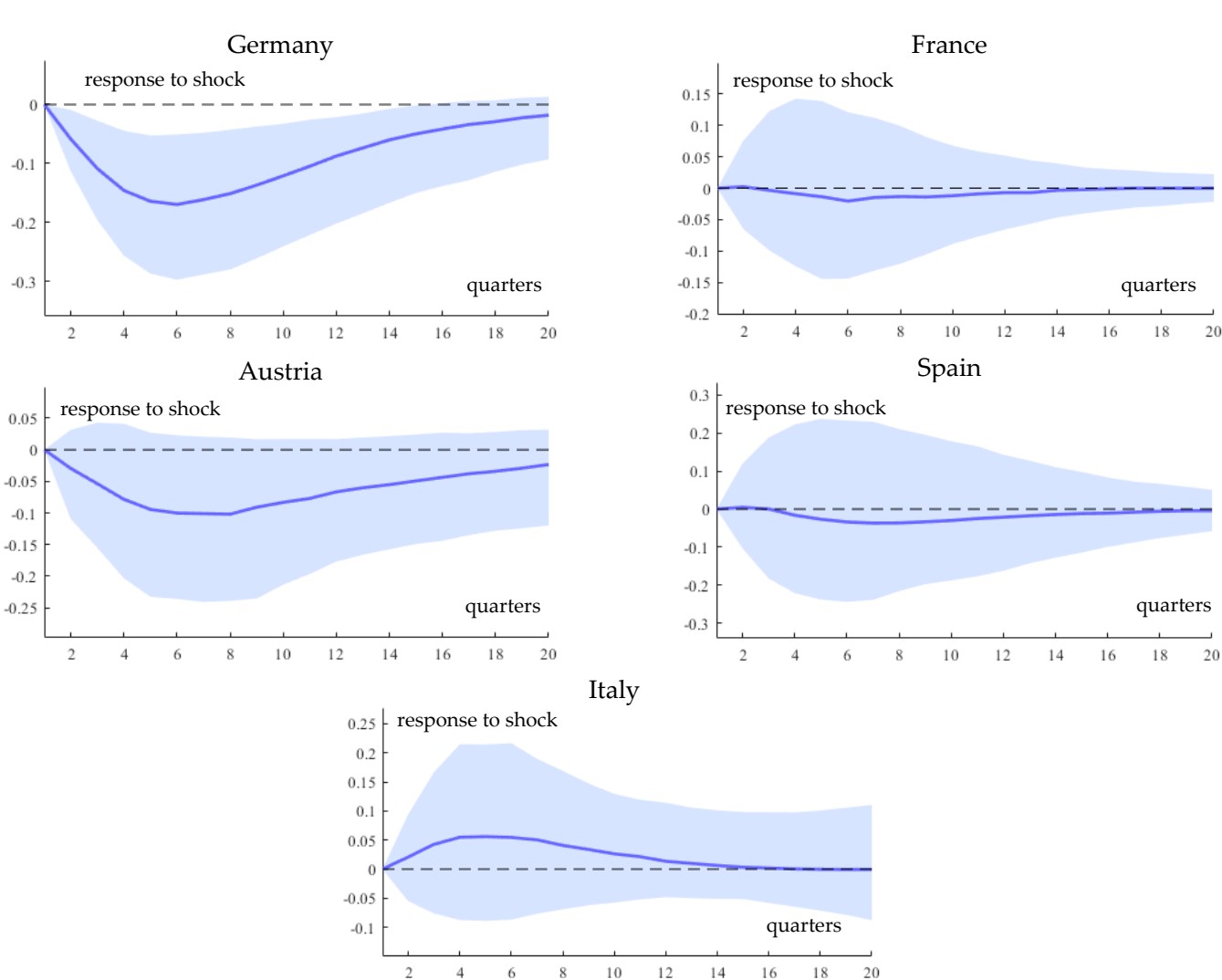

**Figure A1.** Dynamics of the economic cycle as a result of the shock from the level of loans granted to non-financial companies in the states of the central-western region of Europe. Note: the line represents the response of the economic cycle due to the shock, and the highlighted area represents the 95% confidence interval. Source of the data: own processing based on data provided by Eurostat and ECB.

**Appendix B**

Romania

Czechia

Poland

Hungary

Bulgaria

**Figure A2.** Dynamics of the economic cycle as a result of the shock at the level of credits granted to the population in the CEE states. Note: the line represents the response of the economic cycle due to the shock, and the highlighted area represents the 95% confidence interval. Source of the data: own processing based on data provided by Eurostat and ECB.

**Appendix C**

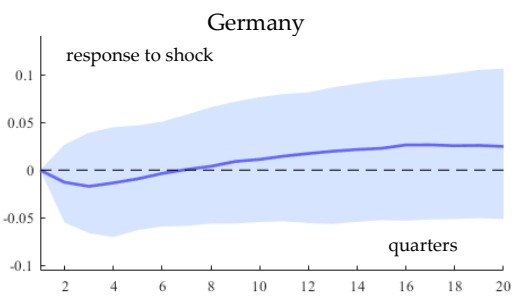

Germany

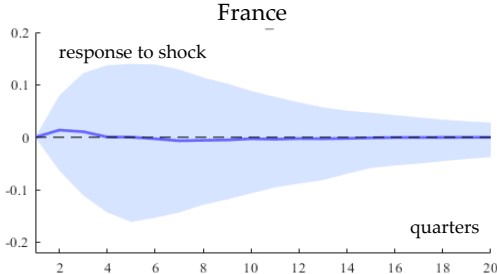

France

**Figure A3.** *Cont.*

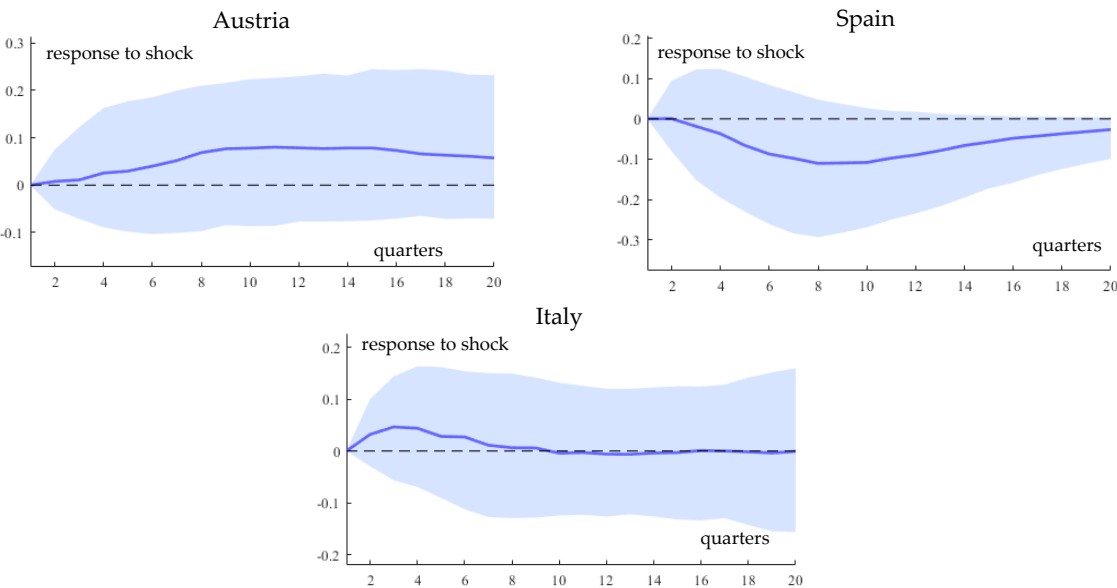

**Figure A3.** Dynamics of the economic cycle as a result of the shock on the level of credits granted to the population in the states of the central-western region of Europe. Note: the line represents the response of the economic cycle due to the shock, and the highlighted area represents the 95% confidence interval. Source of the data: own processing based on data provided by Eurostat and ECB.

## Appendix D

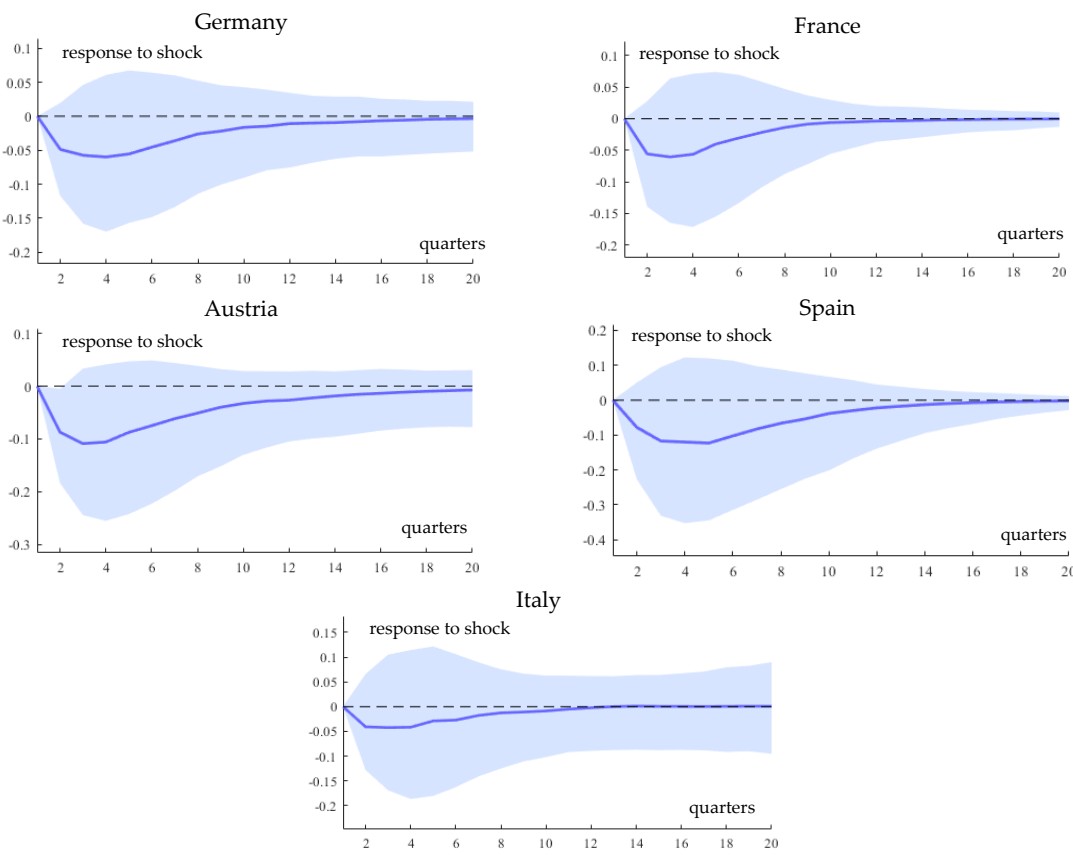

**Figure A4.** Dynamics of the economic cycle as a result of the shock at the CLIFS level in the states of the central-western region of Europe. Note: the line represents the response of the economic cycle due to the shock, and the highlighted area represents the 95% confidence interval. Source of the data: own processing based on data provided by Eurostat and ECB.

**Appendix E**

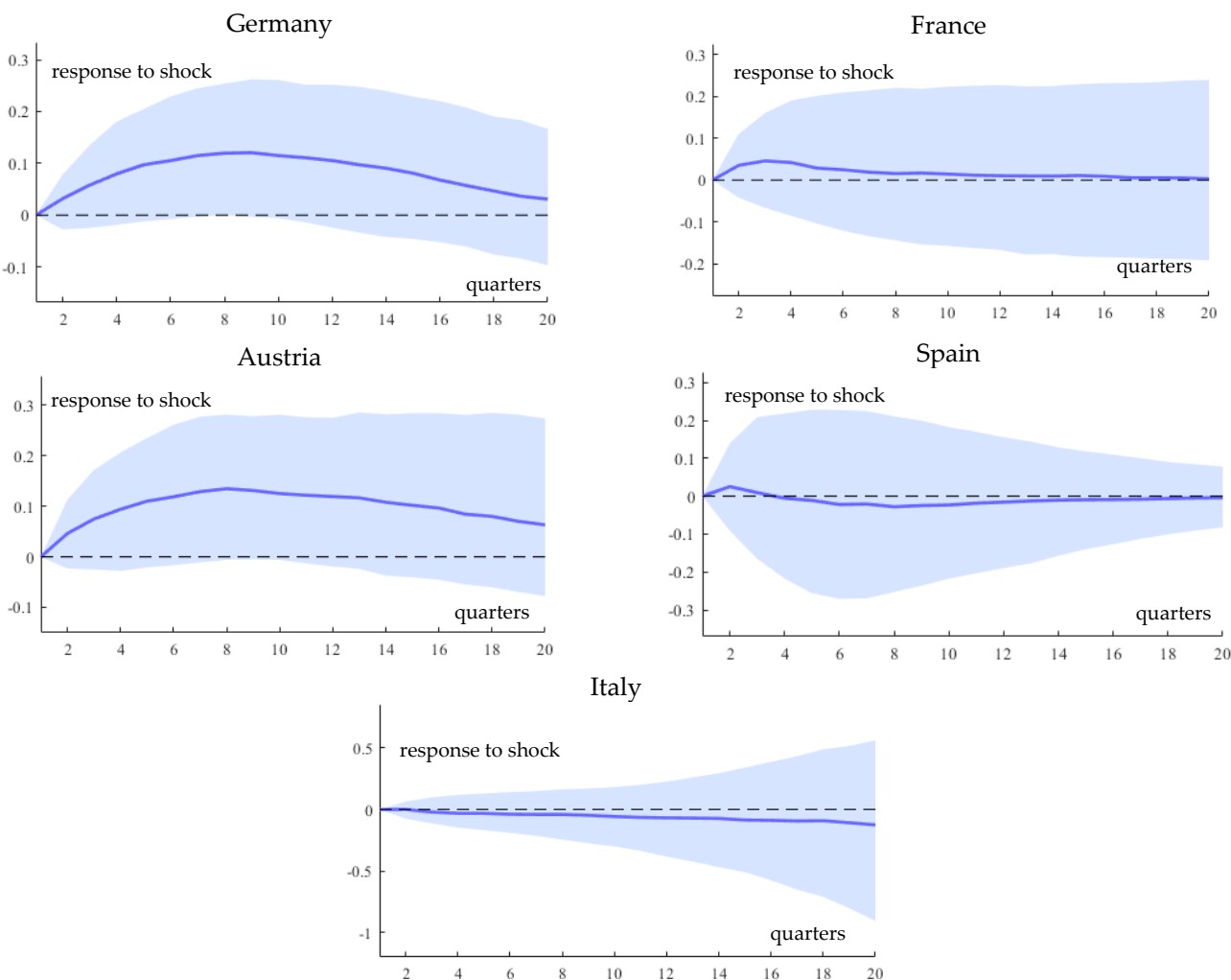

**Figure A5.** Dynamics of the economic cycle as a result of the real estate price shock in the states of the cCentral-wWestern region of Europe. Note: the line represents the response of the economic cycle due to the shock, and the highlighted area represents the 95% confidence interval. Source of the data: own processing based on data provided by Eurostat and ECB.

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
