# Peer review of "Impact of Financial Factors on the Economic Cycle Dynamics in Selected European Countries"

_jrfm, doi:10.3390/jrfm16120492_

Round 1

Reviewer 1 Report

Comments and Suggestions for Authors

Review of the paper : Impact of financial factors on the business cycle dynamics in se- 2 lected European countries

I would like to congratulate the authors on the topic chosen and the work presented.

From my point of view, I would like to make a few comments in order to improve the paper.

In the abstract, I believe the authors should present the study's main contributions to the literature and the evolution of the topic, as well as the research, practical and social implications (some of which are described in the conclusion). Providing a more robust, complete and concise abstract.

In the introduction, which is a vital part of any work, the authors should present more clearly and robustly the relevance and differentiation of the work presented, as well as the implications.

The methodology, which is what allows conclusions to be drawn and results to be obtained, I believe is correct, among other things, in order to achieve the proposed objectives. However, I believe that the authors should describe the robustness tests carried out on the model more clearly.

The discussion of the results is clear and understandable, and the authors can make a connection with the results / work presented in the literature review. The conclusion is presented in a clear and comprehensible way. At this point, the authors should present future avenues for research as well as limitations / difficulties encountered in carrying out the work.

On the last point, the literature review, I think that the authors could have gone further, both in terms of the work presented and in terms of how up-to-date it is; out of 19 references, only five are from the last five years. Current references are synonymous with the topicality of the subject, and I think the authors should try to achieve a ratio of no less than 30%, works from the last 5 years, as opposed to the 26% presented.

Reviewer 2 Report

Comments and Suggestions for Authors

The study assesses the impact of various shocks (recorded in the banking, financial and real estate markets) on the economic cycle. Two datasets are considered: for five Central and Eastern European countries and for five Central and Western European countries. Based on individual (country-level) and group (region-level) analyses, the paper provides additional evidence on the dynamics of economic cycle under the influence of shocks in banking, financial and real estate markets.

From the point of view of the content (especially the empirical analyses), the study may be of interest to researchers and practitioners, but needs some adjustments regarding the literature review and reporting of results. For more details, see the comments below.

 1. The title must be in accordance with the content of the article. The title refers to "business cycle dynamics", and in the article analyzes "dynamics of economic cycle".

2. The abstract needs a background. The first idea in the abstract should place the debate in a wider context and highlight the research question undertaken by the authors. Also, the abstract can be improved by pointing out the following aspects: period of analysis, sampling technique and practical implications of the study.

3. The first section must have a title (Introduction). This section (Introduction) should integrate ideas about all three types of crises. The integration of all variables (associated with the banking sector, the capital market and the real estate market) in the broad spectrum to financial factors is not enough. At the same time, according to the instructions for the authors, the introduction must highlight the purpose of the paper, assess the current state of the research field and, if any, the controversial and divergent hypotheses. The three bibliographic sources are relevant for research, but not sufficient to capture the current state of the research field.

4. The literature review should begin with a synthesis idea that captures the results of previous studies. The results of previous studies should not be assumed (taken over) by the authors (e.g.: "if we quantify the lagged effect of this shock" - lines 112-113).

The authors must provide additional details on cited studies. The synthesis of the results of previous studies is sterile if the analyzed variables are not also pointed out. Convergence / divergence at the level of results must also be argued from the perspective of the analyzed variables. At the same time, the authors need to assess of the current state of knowledge by referring to more recent studies. Starting from the results of the literature review, the authors can formulate the hypotheses that underlie their research.

5. Research methodology needs more clarity. Authors must state whether all variables have the same reference base (they are monthly/quarterly/annual values). In the current version, it is deduced that some data are quarterly and others are annual / monthly (lines 136-147; e.g.: "annualized quarterly growth rate"). This aspect is clarified only in the final part of the section, when the variables of the regression model are detailed. At the same time, authors must provide additional details on the dependent variable (e.g.: measurement indicators & data series for economic cycle dynamics).

6. In the 4th section (Results and discussions) some technical details are required. Specifically, for graphical representations in Figures 1-3 (and Appendixes A-D) the represented variables must be defined (the axes of the graphs - Ox, Oy, the curves and the colored areas).

In interpreting the results highlighted in Figure 1, the authors must explain the particular case of Hungary (in addition to the ideas in lines 217-220). Similarly, authors must explain the particular cases in Figures 2-3 and Appendices A-D.

7. The section of conclusions should begin with a synthesis on the results of the literature review on the topic under investigation. Then, empirical research results need to be refined. In the current version, in the conclusions section are taken up almost identically ideas from the previous sections.

8. The authors must point out the limits of their own research.

9. Abbreviations must be explained at the first use (e.g.: DSGE, VAR, CEE).

Reviewer 3 Report

Comments and Suggestions for Authors

These results support the general intuition commonly known. It is worth publishing the article as another example of empirically supported documentation of the impacts demonstrated by the author.

Author Response

We are very grateful for your positive feed-back. We have made additional improvements to our paper based on the comments of the reviewers.

Round 2

Reviewer 2 Report

Comments and Suggestions for Authors

I appreciate the fact that the authors took into account the recommendations made.

I congratulate the authors for the responsibility with which they revised the manuscript.

In the revised form, the article will be more accessible to the public.